# Electronic Health Literacy, Psychological Distress, and Quality of Life in Urological Cancer Patients: A Longitudinal Study During Transition from Inpatient to Outpatient Care

**DOI:** 10.3390/curroncol32110637

**Published:** 2025-11-13

**Authors:** Dominik Fugmann, Steffen Holsteg, Ralf Schäfer, Günter Niegisch, Ulrike Dinger, André Karger

**Affiliations:** 1Clinical Institute of Psychosomatic Medicine and Psychotherapy, Medical Faculty and University Hospital Düsseldorf, Heinrich-Heine-University Düsseldorf, Centre for Integrated Oncology Düsseldorf (CIO ABCD), 40225 Düsseldorf, Germany; steffen.holsteg@med.uni-duesseldorf.de (S.H.); ralf.schaefer@med.uni-duesseldorf.de (R.S.); ulrike.dinger-ehrenthal@med.uni-duesseldorf.de (U.D.); andre.karger@med.uni-duesseldorf.de (A.K.); 2Centre for Integrated Oncology Aachen Bonn Cologne Düsseldorf (CIO ABCD), 40225 Düsseldorf, Germany; 3Department of Psychosomatic Medicine and Psychotherapy, LVR Hospital Düsseldorf, Clinic of the Heinrich-Heine-University Düsseldorf, 40629 Düsseldorf, Germany; 4Department of Urology, Medical Faculty and University Hospital Düsseldorf, Heinrich-Heine-University Düsseldorf, Centre for Integrated Oncology Düsseldorf (CIO ABCD), 40225 Düsseldorf, Germany

**Keywords:** eHealth literacy, internet-based intervention, urology, prostate neoplasms, kidney neoplasms, urinary bladder neoplasms, testicular neoplasms, psycho-oncology, psychological distress, quality of life

## Abstract

Urological cancers, such as prostate, bladder, kidney, and testicular cancer, are common and often accompanied by a high emotional burden. Many patients do not receive enough psychological support, especially after leaving hospital care. The internet could help patients find useful information and support services, but this requires skills to search, understand, and use health information online, known as electronic health literacy. In this study, we compared patients with higher and lower levels of electronic health literacy during hospital treatment and three months after discharge. Initially, patients with higher skills reported less distress, fewer symptoms of depression, and better quality of life. However, these differences disappeared once age was taken into account, as older patients tended to have lower digital skills. These findings suggest that digital support tools may not reach those who could benefit most. Future services should be tailored to older patients’ needs to ensure equitable access.

## 1. Introduction

Urological cancers are among the most common types of cancer worldwide [1]. A diagnosis of urological cancer is frequently associated with a decline in quality of life [2]. Correspondingly, psychological symptoms are common, although there are specific differences depending on the type of urological cancer and its treatment [3]. Notably, however, only 13% of patients with penile or prostate cancer and 12% of patients with other urological cancers receive psychological support—compared to 28% of cancer patients overall [4].

In Germany, screening for psychosocial distress of all cancer patients is mandatory in cancer centres certified by the German Cancer Society (Deutsche Krebsgesellschaft). If elevated distress is identified, or if a patient requests support, psycho-oncological support is offered [5]. However, due to limited capacity and other contributing factors, not all patients receive either the screening or the counselling [6]. This is especially problematic after the completion of active treatment, when many patients lose contact with psychosocial support systems—often due to a lack of awareness about available resources [7].

High eHL may help to close this care gap. According to Norman and Skinner [8], eHL is defined as: “the ability to seek, find, understand, and appraise health information from electronic sources and apply the knowledge gained to addressing or solving a health problem.” Based on this definition, cancer patients with high eHL may be able to independently locate and interpret information about available support services [9], and adjust their health behaviour accordingly—including the identification and use of (online) psychosocial support.

Whether high eHL actually contributes to improved quality of life or reduced psychological symptoms remains unclear. A meta-review comprising ten systematic reviews found that eHealth can enhance perceived support, knowledge, and cancer-related awareness. However, the findings regarding psychological symptoms and quality of life were inconsistent [10]. This pattern is echoed in more recent reviews and individual studies [11,12]. In breast cancer, systematic reviews of eHealth interventions indicate that such approaches can improve quality of life, self-efficacy, and reduce psychological distress and fatigue [13]. In the context of urological cancers, the evidence is less clear. A review on eHL in prostate cancer highlighted that few studies exist, and that the construct of eHL is often poorly operationalised [14]. Evidence suggests that limited health literacy among prostate cancer patients is associated with heightened psychological distress; however, no significant association has been found with physical well-being [15]. One study suggests that higher eHL in prostate cancer patients is associated with better global quality of life and improved emotional functioning [16]. For urological cancers, systematic reviews suggest that technology-based supportive care interventions may improve psychological outcomes and quality of life in prostate cancer, although the evidence remains limited and heterogeneous [17,18]. 

The present study aims to investigate whether urological cancer patients with high levels of eHL (HH) exhibit fewer psychological symptoms (depression, anxiety, distress) and higher quality of life during the transition phase from inpatient to outpatient care, compared to the group with low eHL (LH).

## 2. Materials and Methods

The data for this study were drawn from the same sample as previously reported in Fugmann et al. [19], though re-analysed here to address a distinct research question.

Health literacy is influenced by various factors such as age, education, and gender [20,21]. Its impact on health behaviour is mediated by attitudes towards health behaviour, disease knowledge, and self-efficacy [22,23]. Health behaviour, in turn, has a direct effect on health outcomes such as psychological symptoms or quality of life (see Figure 1).

### 2.1. Study Design and Participants

A prospective, single-centre observational study was conducted over a one-year recruitment period (October 2020 to October 2021). Inclusion criteria were assessed via an initial screening questionnaire (T0), focusing on patients with urological cancers (diagnoses according to ICD-10 codes: C60, C61, C62, C63, C64, C65, C66, C67, C68) who were receiving inpatient care at the urology clinic of the Comprehensive Cancer Centre Düsseldorf. Several times a week, a trained study staff member approached all hospitalized urological cancer patients directly and reviewed the screening questionnaire together with them in a face-to-face conversation to determine eligibility. Participants had to be 18 years or older, have a score of ≥5 on the Distress Thermometer (DT) [24] and/or express a desire for support. This approach aligned with the primary study’s aim to include only distressed patients in order to examine factors predicting subsequent use of psycho-oncological support. Additionally, participants were required to have the legal capacity to consent, sufficient proficiency in German, and an expected survival time of more than six months. The T0 questionnaire also recorded demographic data, including age, gender, and educational level. 

Patients who met inclusion criteria were given comprehensive information about the study and provided written informed consent. Following enrolment, participants completed the T1 questionnaire and received at least one psycho-oncological consultation during their inpatient stay, with up to five follow-up outpatient consultations. Three months later, the T2 questionnaire was mailed to participants by postal delivery (see Figure 2). All questionnaires were completed manually using pen and paper. The study protocol was retrospectively registered with the German Clinical Trials Registry (No. DRKS00024012, registration date: 11 January 2021) [25], and ethical approval was obtained from the Ethics Committee of the Medical Faculty at Heinrich Heine University, Düsseldorf, Germany (approval code: No. 2020-1085; approval date: 7 September 2020). The study was funded by the Heigl Foundation, Düsseldorf, Germany.

### 2.2. Instruments

At baseline (T1), participants provided sociodemographic data and clinical background information, including cancer diagnosis, metastatic status, and recent treatments within the preceding two months. Information on previous psychiatric or psychological care was collected using items adapted from the PO-BADO scale [26]. Functional status was determined using the Eastern Cooperative Oncology Group (ECOG) performance scale, which classifies activity limitations on a five-point scale ranging from 0 (fully active) to 4 (completely disabled) [27]. To evaluate eHealth literacy, participants completed the electronic Health Literacy Scale (eHEALS), an eight-item instrument capturing perceived ability to locate, interpret, and apply health-related information on the internet [8]. Each item is rated from 1 (“strongly disagree”) to 5 (“strongly agree”), yielding a total score between 8 and 40. Scores of 26 or higher are typically interpreted as reflecting sufficient digital health literacy [8]. Psychological distress was measured using the Distress Thermometer (DT), a single-item 0–10 scale where values above 5 denote clinically relevant distress [24]. Anxiety and depressive symptoms were captured via the Patient Health Questionnaire-4 (PHQ-4), composed of two subscales: the GAD-2 for anxiety and PHQ-2 for depression. Thresholds of ≥4 on the PHQ-4 or ≥3 on either subscale indicate elevated symptom levels [28,29,30]. Health-related quality of life was assessed using the EORTC QLQ-C30, which evaluates multiple dimensions of functioning (physical, emotional, cognitive, role, and social) as well as overall health and symptom burden. Responses are scored on a four-point Likert scale and transformed to a 0–100 metric for cross-domain comparison [31]. At follow-up (T2), participants again completed the DT, PHQ-4, and QLQ-C30 to monitor psychosocial changes over time.

### 2.3. Statistical Analysis

For age, group differences between LH and HH were assessed using the Wilcoxon rank-sum test due to a non-normal distribution. Associations between categorical variables—such as relationship status, sex, level of education, cancer type, metastatic disease, treatments administered within the previous two months, functional performance, prior psychological or psychiatric care, initial request for psycho-oncological support (T0), and subsequent use of psycho-oncological follow-up—were examined using chi-squared (χ^2^) tests. Group differences between LH and HH regarding differences in psychological symptoms (PHQ-2, GAD-2, DT), and quality of life (domains of the EORC QLQ-C30) at T1 and T2 were assessed via Wilcoxon rank sum test due to a non-normal distribution. If data were missing, the dataset was removed from the respective analysis. No imputation methods were applied. In a second step, group differences were adjusted for age using residuals. To examine whether age moderated the relationship between eHL (eHEALS) and various domains of health-related quality of life and psychological symptoms, separate multiple linear regression models were conducted for each outcome variable, including an interaction term (dichotomised eHEALS × Age). Statistical analyses were performed using R (version 4.4.1, R Core Team, 2019), with the package “psych” [32]. This study represents a secondary analysis of data from a prospective observational study originally designed to identify factors predicting the use of psycho-oncological support. As no formal power calculation was conducted for the parent study, we performed a post hoc power analysis for the present analysis, which indicated a power of 0.78 to detect medium-sized effects (Cohen’s f^2^ = 0.15, α = 0.05). Therefore, the findings should be interpreted as exploratory.

## 3. Results

### 3.1. Sample and Dataset Generation

Of n = 627 urological cancer patients in the recruitment period, n = 501 were screened for inclusion criteria. N = 139 patients met the criteria and consented to participate in the study. Completed T1 questionnaires were received from n = 108 patients; n = 71 patients returned completed T2 questionnaires (see Figure 3).

### 3.2. Sample Characteristics

The socio-demographic characteristics and disease-specific data of the sample are summarised in Table 1.

The recruited patients had a mean age of 58.53 years (*SD* = 17.16). The sample consisted of 89% men, and 71% were in a relationship. Approximately half of the participants had 10 years or less of formal education, while the other half had more than 10 years. The most common cancer types were prostate cancer (31%), bladder cancer (24%), testicular cancer (23%), and kidney cancer (19%). Metastatic disease was present in 35% of the patients. Surgical treatment within the past two months had been performed in 65%, and 27% had received chemotherapy. 73% showed only minor impairment in their performance status. 77% expressed a need for psychosocial support, and 22% made use of psycho-oncological follow-up services after the mandatory initial consultation. There were no significant differences between the low and high digital health literacy groups in any of the assessed variables, except for age: LH: *M* = 64.17, *SD* = 15.88; HL: *M* = 56.14, *SD* = 17.09; *p* = 0.019 (see Table 1). 

### 3.3. Group Differences by Electronic Health Literacy

At T1, participants with higher digital health literacy reported significantly lower distress (*p* = 0.047, *r* = 0.19), fewer depressive symptoms (*p* = 0.004, *r* = 0.28), better physical functioning (*p* = 0.001, *r* = 0.31), better role functioning (*p* = 0.013, *r* = 0.24), and better emotional functioning (*p* = 0.002, *r* = 0.30), as indicated by Wilcoxon rank-sum tests. At T2, significant differences between high and low literacy groups were found for physical functioning (*p* < 0.001, *r* = 0.32), and quality of life (*p* = 0.003, *r* = 0.29). However, none of these associations remained statistically significant after adjusting for age using residualised values (see Table 2). 

### 3.4. Age as Moderator

Across all domains at both T1 and T2 (distress, depression, anxiety, physical, role, cognitive, emotional, and social functioning, as well as global quality of life), no significant interaction effects were observed. These findings suggest that age did not moderate the association between eHL and any of the assessed domains of quality of life or psychological symptoms (see Table 3).

## 4. Discussion

This secondary analysis examined whether differences exist between individuals with low (LH) and high (HH) levels of eHL in relation to psychological symptoms (distress, anxiety, and depression) and health-related quality of life (physical, role, cognitive, emotional, social functioning, and global quality of life) during inpatient cancer treatment and at a follow-up assessment three months after the initial data collection.

Group differences identified in the primary analysis were no longer detectable after adjusting for age. Age did not moderate the relationship between eHL and psychological symptoms or quality of life domains. These findings align with inconsistent evidence in the broader cancer literature [10,11,12], but contrast with results in breast cancer populations, where moderation effects have been observed [13]. Evidence from breast cancer studies on digital interventions is mixed, likely due to differences in intervention type, patient age, and disease context. Younger, digitally literate breast cancer patients differ from older urological patients, who face greater usability and access barriers [33,34,35]. The variability in outcomes probably reflects these contextual differences rather than truly inconsistent effects. This may suggest that existing online health resources for breast cancer patients are either better tailored to patient needs or that this group is generally more responsive to digital support tools.

To date, robust systematic reviews focusing specifically on urological cancers remain scarce. One study among prostate cancer patients reported associations between higher eHL and improved emotional functioning and global quality of life [16]. However, that study also identified a significant age difference between high and low eHL groups, without applying statistical adjustment for age. Thus, observed effects may have been driven by age differences, similar to our initial analysis.

Consistent with previous literature [33,34], our findings indicate that older cancer patients are less familiar with, and less inclined to use, digital health resources. Older age appears to be a potential barrier to accessing digital health tools [19]. Compared to a general population survey from Germany (average eHEALS 31 points) [35], our sample showed lower overall scores (M = 26.9). While the high eHL subgroup was comparable to population averages, the low eHL subgroup scored markedly below. This highlights substantial heterogeneity in patients’ digital health skills.

The timing of our assessment may also explain the lack of observed associations: patients in the acute phase of cancer treatment (inpatient and shortly after discharge) are likely to prioritise physical symptoms and immediate recovery [36]. Moreover, high psychological distress may impair the ability to search for and process online health information [37]. During hospitalisation, access to professional support is immediate, which may reduce the perceived need for digital resources. Local factors, such as poor internet access or lack of privacy, may also limit digital engagement in inpatient settings. It is therefore conceivable that the influence of eHL on psychological burden and quality of life may become more apparent later in the treatment trajectory.

Alternatively, available digital health resources for urological cancer patients may be insufficiently tailored to their needs. It is also possible that a subset of patients prefer face-to-face interactions—although existing evidence tends to suggest the opposite [19].

### Limitations

This study has several limitations. First, it employed a single-centre convenience sample, which reduces generalisability. Second, the relatively small sample size, particularly at follow-up, limits statistical power for subgroup or moderator analyses. Third, the analysis did not include certain potentially relevant moderators, such as self-efficacy or disease knowledge [22,23]. Fourth, the observational design precludes causal inferences. Finally, while our findings provide valuable insights into the association between age, health literacy, and digital engagement in urological cancer, a major methodological limitation lies in the use of the original eHEALS, which primarily assesses Web 1.0 skills such as searching for and appraising online health information. This tool does not adequately reflect contemporary digital behaviour, where patients increasingly interact with online communities, social media, and AI-based health applications. More comprehensive instruments such as the Digital Health Literacy Instrument (DHLI), which extends to Web 2.0 skills including interactive communication and content sharing [38], or newer Web 3.0-based measures that incorporate competencies in navigating algorithm-driven, participatory, and AI-enhanced environments [39,40], could provide a more accurate representation of modern eHealth literacy. Future research should therefore adopt such updated instruments to capture patients’ ability to critically evaluate and use AI-supported or social-media-mediated health information, which may substantially influence engagement and self-management in cancer care. Using a more comprehensive eHealth literacy instrument that includes Web 2.0 and 3.0 competencies might have revealed smaller age-related gaps in digital engagement, as recent evidence shows that an increasing proportion of older adults actively use social media for health information while differing in their ability to evaluate and manage misinformation online [41,42].

## 5. Conclusions

These findings suggest that, in this sample, eHL was not significantly associated with psychological outcomes or quality of life during the transition from inpatient to outpatient care. However, higher age was linked to lower eHL, which could still limit the ability of older patients to access and benefit from digital health resources. Therefore, age-sensitive strategies and targeted guidance may help ensure equitable access to such resources. In addition, the design of digital support services should be closely aligned with the specific informational and emotional needs of patients with urological cancers to maximise their relevance and potential impact.

## Figures and Tables

**Figure 1 curroncol-32-00637-f001:**
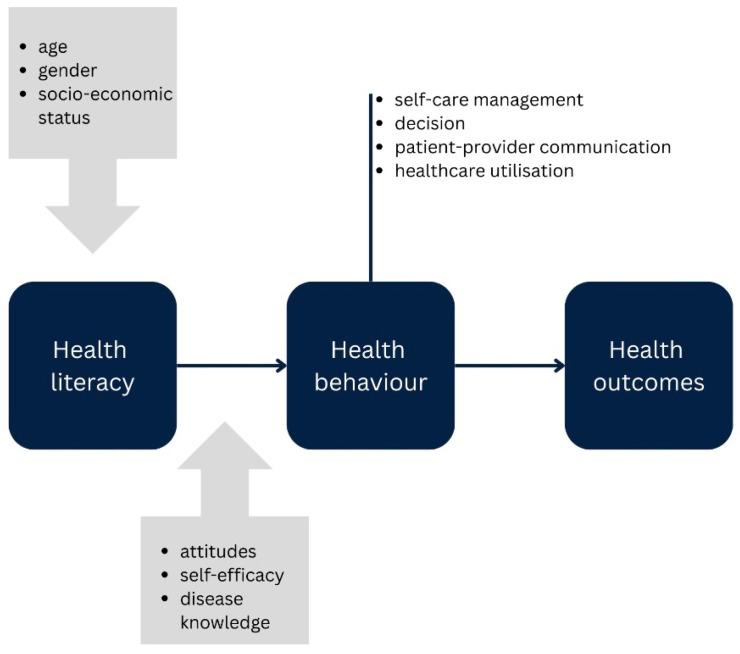
Conceptual framework.

**Figure 2 curroncol-32-00637-f002:**
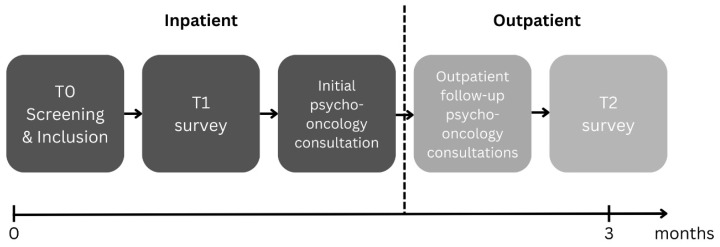
Procedure.

**Figure 3 curroncol-32-00637-f003:**
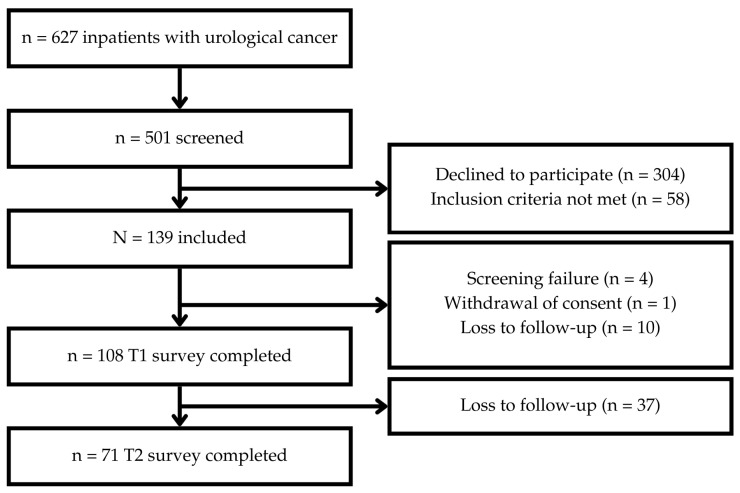
Flow Chart.

**Table 1 curroncol-32-00637-t001:** Sample description.

		Total (n = 108)		LH (n = 35)		HH (n = 72)		Between Groups			
**Variable**	Levels	M	SD	M	SD	M	SD		W	*p*	
**Age**	Years	58.53	17.16	64.17	15.88	56.14	17.09		907	0.019	*
**eHealth literacy**	eHEALS score	26.95	7.95	17.51	5.39	31.58	3.89				
		N	%	N	%	N	%	χ^2^	df	*p*	
**Partner**	Yes	77	71.30	25	71.43	52	72.22				
	No	30	27.78	9	25.71	20	27.78				
	Unknown	1	0.93	1	2.86	0	0			0.437	
**Sex**	Male	96	88.89	28	80.00	67	93.06				
	Female	12	11.11	7	20.00	5	6.94			0.056	
**Education**	≤10 years	54	50.00	23	65.71	31	43.06				
	>10 years	50	46.30	12	34.29	37	51.39				
	n.a.	4	3.70	0	0	4	5.56	2989	1	0.084	
**Cancer type**	Prostate	33	30.56	10	28.57	23	31.94				
	Bladder	26	24.07	13	37.14	13	18.06				
	Penis	4	3.70	0	0	4	5.56				
	Testicular	25	23.15	5	14.29	19	26.39				
	Kidney	20	18.52	7	20.00	13	18.06			0.149	
**Metastases**	Yes	38	35.19	12	34.29	25	34.72				
	No	54	50.00	15	42.86	39	54.17				
	Unknown	15	13.89	8	22.86	7	9.72				
	n.a.	1	0.93	0	0	1	1.39			0.186	
**Treatments in the** **past two months**	Operation	70	64.81	24	68.57	45	62.50	0.160	1	0.689	
	Chemotherapy	29	26.58	9	25.71	19	26.39	<0.001	1	1	
	Radiotherapy	4	3.70	1	2.86	3	4.17			1	
	Hormone therapy	3	2.78	0	0	3	4.17			0.549	
	None	20	18.52	6	17.14	14	19.44	<0.001	1	0.982	
**Performance status ^†^**	Low impairment	79	73.15	22	62.86	56	77.78				
	High impairment	26	24.07	13	37.14	13	18.06				
	n.a.	3	2.78	0	0	3	4.17	3230	1	0.072	
**History of psychological** **or psychiatric treatment**	Yes	33	30.56	10	28.57	23	31.94				
	No	73	67.59	25	71.43	47	65.28				
	n.a.	2	1.85	0	0	2	2.78	0.050	1	0.824	
**Request for support** **(T0)**	Yes	83	76.85	28	80.00	54	75.00				
	No	13	12.04	3	8.57	10	13.89				
	Not Specified	11	10.19	3	8.57	8	11.11				
	n.a.	1	0.93	1	2.56	0	0			0.717	
**Use of follow-up psycho-oncology consultations**	Yes	24	22.22	10	28.57	14	19.44				
	No	84	77.78	25	71.43	58	80.56	0.664	1	0.415	

*M* = Mean. *SD* = Standard deviation. *W* = Wilcoxon rank sum statistics. χ^2^ = Chi square statistics. Df = Degrees of freedom from χ^2^ test. The sample consisted of 108 patients (n = 108). Information on age, sex, education, and requests for support was collected during the screening for study inclusion (T0). All remaining variables were gathered at T1, prior to the initial psycho-oncology consultation during inpatient care. Group differences were tested using the Wilcoxon rank-sum test for age. Categorical variables—including education level, recent treatments (surgery, chemotherapy, or none), performance status, prior psychological or psychiatric care, and utilisation of psycho-oncological follow-up—were analysed with χ^2^ tests. For variables with small expected frequencies, such as partner status, sex, cancer type, metastatic status, recent radiotherapy or hormone therapy, and initial request for support, Fisher’s exact test was applied. † Performance status was classified as low impairment (ECOG 0–2) or high impairment (ECOG 3–4). *p* < 0.05 *, *p* < 0.01 **, *p* < 0.001 ***.

**Table 2 curroncol-32-00637-t002:** Comparison of low (LH) and high eHL (HH) groups regarding psychological symptoms and quality of life.

						Group Comparison (Wilcoxon Rank Sum Test)									Group Comparison (Age-Adjusted via Residuals)						
			Total			LH		HH			Between Groups				LH		HH		Between Groups		
	Variable	N	M	SD	N	M	SD	N	M	SD	W	*p*		r	SD	MD	SD	MD	W	*p*	r
**T1**	Distress Thermometer	107	6.61	2.55	35	7.26	2.41	71	6.35	2.55	1536	0.047	*	0.19	2.38	0.64	2.55	0.64	1279	0.809	0.02
	Depression (PHQ-2)	108	2.23	1.87	35	3.00	1.93	72	1.89	1.73	1683	0.004	**	0.28	1.92	−0.05	1.72	<−0.01	1290	0.845	0.02
	Anxiety (GAD-2)	108	2.12	1.60	35	2.29	1.51	72	2.06	1.66	1401	0.342		0.09	1.48	0.01	1.64	−0.18	1252	0.960	<0.01
	Physical functioning	103	71.07	30.42	35	56.76	31.53	68	78.43	27.24	742	0.001	**	0.31	29.57	0.12	27.04	15.74	1204	0.925	0.01
	Role functioning	106	64.15	39.29	35	50.95	36.81	71	70.66	39.08	890	0.013	*	0.24	36.81	−16.97	38.80	23.31	1218	0.872	0.02
	Cognitive functioning	107	74.61	27.22	35	61.90	29.31	72	80.79	24.01	778	<0.001	***	0.32	28.93	3.82	23.95	4.71	1206	0.722	0.03
	Emotional functioning	105	52.78	25.23	35	42.14	16.78	70	58.10	27.10	769	0.002	**	0.30	16.76	−0.86	27.06	3.27	1270	0.762	0.03
	Social functioning	107	69.16	30.18	35	61.90	30.94	72	72.69	29.38	989	0.064		0.18	28.98	5.86	29.28	7.60	1253	0.966	<0.01
	Quality of life	107	47.98	28.66	35	36.19	28.72	72	53.70	27.01	812	0.003	**	0.29	27.85	−12.08	26.95	2.15	1343	0.584	0.05
**T2**	Distress Thermometer	62	4.23	2.58	14	4.14	1.75	48	4.25	2.79	341	0.939		0.01	1.69	−0.08	2.79	−0.70	327	0.886	0.02
	Depression (PHQ-2)	68	1.65	1.31	16	1.88	1.26	52	1.58	1.33	477	0.370		0.09	1.23	−0.14	1.33	0.32	437	0.767	0.04
	Anxiety (GAD-2)	69	1.41	1.45	17	1.53	1.50	52	1.37	1.44	478	0.613		0.05	1.47	−0.48	1.41	−0.27	437	0.950	<0.01
	Physical functioning	71	74.37	26.13	17	58.82	28.21	54	79.26	23.66	263	0.007	**	0.26	26.99	2.13	22.65	8.40	475	0.835	0.02
	Role functioning	70	65.48	30.47	17	53.92	31.47	53	69.18	29.49	319	0.066		0.18	31.45	−2.66	28.29	2.58	452	0.989	<0.01
	Cognitive functioning	68	73.77	28.55	16	68.75	31.55	52	75.32	27.71	369	0.483		0.07	31.43	7.20	27.66	6.31	368	0.492	0.08
	Emotional functioning	67	65.17	27.75	17	61.76	24.48	50	66.33	28.91	366	0.397		0.08	23.26	3.05	28.91	8.72	452	0.703	0.05
	Social functioning	66	75.25	28.68	16	62.50	34.69	50	79.33	25.55	290	0.085		0.17	34.25	9.68	25.10	3.88	388	0.863	0.02
	Quality of life	69	61.35	23.04	17	52.45	19.93	52	64.26	23.42	296	0.041	*	0.20	19.26	−0.36	21.76	0.22	456	0.851	0.02

Physical functioning, role functioning, cognitive functioning, emotional functioning, social functioning, and quality of life were assessed with the EORTC QLQ-C30 questionnaire. T1 = immediately after inclusion in the study, T2 = 3 months after inclusion. *M* = Mean. *SD* = Standard deviation. *MD* = Median. *W* = Wilcoxon rank sum statistics. *r* = Rank correlation coefficient (effect size). *p* < 0.05 *, *p* < 0.01 **, *p* < 0.001 ***.

**Table 3 curroncol-32-00637-t003:** Moderation analysis of eHL (eHEALS) and age on quality of life and psychological symptoms.

	Domain	Main Effect eHEALS	Main Effect Age	Inter- Action Effect	SE Inter- Action	*p* Value Interaction	CI Lower	CI Upper
**T1**	Distress thermometer	0.79	0	0.02	0.03	0.539	−0.04	0.08
	Depression (PHQ-2)	1.10	0.01	−0.01	0.02	0.585	−0.06	0.03
	Anxiety (GAD-2)	0.36	−0.01	0	0.02	0.910	−0.04	0.04
	Physical functioning	−17.67	−0.19	−0.51	0.36	0.168	−1.22	0.21
	Role functioning	−19.23	−0.27	0.32	0.49	0.521	−0.65	1.29
	Cognitive functioning	−17.03	−0.1	−0.20	0.33	0.551	−0.85	0.45
	Emotional functioning	−16.44	0.09	−0.04	0.32	0.899	−0.66	0.58
	Social functioning	−14.84	0.14	0.54	0.38	0.157	−0.20	1.28
	Quality of life	−14.84	−0.11	−0.34	0.35	0.345	−1.03	0.36
**T2**	Distress thermometer	−0.22	−0.01	0.04	0.05	0.482	−0.07	0.14
	Depression (PHQ-2)	0.40	0	−0.02	0.02	0.389	−0.07	0.03
	Anxiety (GAD-2)	0.35	−0.02	0	0.03	0.943	−0.05	0.05
	Physical functioning	−16.05	−0.40	−0.14	0.44	0.750	−1.01	0.73
	Role functioning	−13.77	−0.49	0.41	0.54	0.454	−0.66	1.48
	Cognitive functioning	−7.51	−0.09	0.27	0.54	0.617	−0.78	1.32
	Emotional functioning	−7.70	−0.03	0.54	0.52	0.306	−0.48	1.56
	Social functioning	−18.67	−0.28	0.63	0.52	0.227	−0.38	1.65
	Quality of life	−8.50	−0.50	0.16	0.40	0.686	−0.61	0.94

Values represent unstandardised regression coefficients, standard errors (SE), *p*-values, and 95% confidence intervals (CI) for the interaction term. Linear regression models were computed for each outcome, including main effects of eHEALS (high vs. low), age (centred), and their interaction. Quality of life domains were assessed using the EORTC QLQ-C30.

## Data Availability

The data that support the findings of this study are available from the corresponding author D.F. upon reasonable request. The data are not publicly available due to ethical and legal restrictions, as the informed consent obtained from participants and the ethics committee approval did not foresee unrestricted data sharing.

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
