# Peer review of "Electronic Health Literacy, Psychological Distress, and Quality of Life in Urological Cancer Patients: A Longitudinal Study During Transition from Inpatient to Outpatient Care"

_curroncol, 2025, doi:10.3390/curroncol32110637_

Round 1
Reviewer 1 Report
Comments and Suggestions for Authors
This paper assesses the important issue of health literacy and age for patients with prostate cancer. The introduction clearly states the gaps in care for this cohort and the potential for digital tools to address it ,
The data are clearly presented and the limitations of the study in terms of sample size are addressed. I feel the discussion could be improved by
- expand the area where the study is compared with others eg breast cancer - why are these studies inconsistent ?
- A major limitation is the use as the authors state of a tool that doesn't reflect todays practice - social media and AI - can they expand this section - how should the next study be developed ? would their conclusions be different if a tool was available that reflected modern day life ?
Author Response
Comment 1: I feel the discussion could be improved by expand the area where the study is compared with others eg breast cancer - why are these studies inconsistent ?
Response 1: Thank you for this valuable suggestion. We have substantially expanded the Discussion to compare our results with prior findings in breast cancer and to clarify why findings across studies may appear inconsistent.
Added text in revised version: Evidence from breast cancer studies on digital interventions is mixed, likely due to differences in intervention type, patient age, and disease context. Younger, digitally lit-erate breast cancer patients differ from older urological patients, who face greater usa-bility and access barriers [33-35]. The variability in outcomes probably reflects these contextual differences rather than truly inconsistent effects.
Comment 2: A major limitation is the use as the authors state of a tool that doesn't reflect todays practice - social media and AI - can they expand this section - how should the next study be developed? would their conclusions be different if a tool was available that reflected modern day life?
Response 2:We fully agree. We revised the Limitations section to elaborate on the limitations of eHEALS and discuss future directions for more contemporary instruments addressing Web 2.0 and AI-based digital health behaviour.
Added text in revised version: Finally, while our findings provide valuable insights into the association between age, health literacy, and digital engagement in urological cancer, a major methodological limitation lies in the use of the original eHEALS, which primari-ly assesses Web 1.0 skills such as searching for and appraising online health infor-mation. This tool does not adequately reflect contemporary digital behaviour, where patients increasingly interact with online communities, social media, and AI-based health applications. More comprehensive instruments such as the Digital Health Liter-acy Instrument (DHLI), which extends to Web 2.0 skills including interactive commu-nication and content sharing [38], or newer Web 3.0-based measures that incorporate competencies in navigating algorithm-driven, participatory, and AI-enhanced envi-ronments [39, 40], could provide a more accurate representation of modern eHealth lit-eracy. Future research should therefore adopt such updated instruments to capture pa-tients’ ability to critically evaluate and use AI-supported or social-media-mediated health information, which may substantially influence engagement and self-management in cancer care. Using a more comprehensive eHealth literacy instru-ment that includes Web 2.0 and 3.0 competencies might have revealed smaller age-related gaps in digital engagement, as recent evidence shows that an increasing proportion of older adults actively use social media for health information while differing in their ability to evaluate and manage misinformation online [40, 41].
Reviewer 2 Report
Comments and Suggestions for Authors
Dear authors and editors,Thank you very much for the opportunity to review this magnificent work.I have just a few comments:
- How was the sample size calculated? Was it sufficient?
- What was the reason for choosing the Distress Thermometer to select the patients?
Good Luck!
Author Response
Comment 1: I have just a few comments: How was the sample size calculated? Was it sufficient?
Response 1: We added a new paragraph to the Statistical Analysis section describing a post hoc power analysis to address the reviewer’s request.
Added text in revised version: This study represents a secondary analysis of data from a prospective observational study originally designed to identify factors predicting the use of psycho-oncological support. As no formal power calculation was conducted for the parent study, we performed a post hoc power analysis for the present analysis, which indicated a power of 0.78 to detect medium-sized effects (Cohen’s f² = 0.15, α = 0.05). Therefore, the findings should be interpreted as exploratory.
Commeent 2: What was the reason for choosing the Distress Thermometer to select the patients?
Response 2: We clarified the rationale for this inclusion criterion in the Study Design and Participants section, linking it to the overarching aim of including distressed patients likely to seek psycho-oncological support.
Added text in revised version: This approach aligned with the primary study’s aim to include only distressed patients in order to examine factors predicting subsequent use of psycho-oncological support.
Round 2
Reviewer 1 Report
Comments and Suggestions for Authors
thank you for considering my comments on the prior version of the manuscript and for the comprehensive nature of the revisions performed in response to peer review